# Residual Matrix Product State for Machine Learning

Ye-Ming Meng[a], Jing Zhang[b], Peng Zhang[b], Chao Gao[a], Shi-Ju Ran[c]

[a]*Department of Physics, Zhejiang Normal University, Jinhua, 321004, China*
[b]*School of Computer Science and Technology, Tianjin University, Tianjin, China*
[c]*Department of Physics, Capital Normal University, Beijing 100048, China*

## Abstract

Tensor network, which originates from quantum physics, is emerging as an efficient tool for classical and quantum machine learning. Nevertheless, there still exists a considerable accuracy gap between tensor network and the sophisticated neural network models for classical machine learning. In this work, we combine the ideas of matrix product state (MPS), the simplest tensor network structure, and residual neural network and propose the residual matrix product state (ResMPS). The ResMPS can be treated as a network where its layers map the "hidden" features to the outputs (e.g., classifications), and the variational parameters of the layers are the functions of the features of the samples (e.g., pixels of images). This is different from neural network, where the layers map feed-forwardly the features to the output. The ResMPS can equip with the non-linear activations and dropout layers, and outperforms the state-of-the-art tensor network models in terms of efficiency, stability, and expression power. Besides, ResMPS is interpretable from the perspective of polynomial expansion, where the factorization and exponential machines naturally emerge. Our work contributes to connecting and hybridizing neural and tensor networks, which is crucial to further enhance our understand of the working mechanisms and improve the performance of both models.

*Keywords:* machine learning, matrix product state, residue network

## 1. Introduction

The tensor network (TN), as a mathematical model that is widely used to describe quantum many-body states [1–4], has been successful applied to machine learning (ML). For instance, TN is used in supervised and unsupervised image classification, natural language processing, etc. [5–11]. Several recent works also demonstrate TN's ability of establishing the connection between physics and artificial intelligence [12, 13]. Nevertheless, despite the high interpretability of TN [14–16], there still exists a considerable performance gap between TN and neural network (NN) [7, 17].

TN itself represents a linear map between quantum states. While in machine learning, TN realizes a non-linear map from the features to the outputs, where there exists a local kernel function [5] that maps the features of the samples to the quantum states in Hilbert space. It is however an open issue to determine whether the NN techniques can enhance TN performance. Several recent works has explored different ways to combine TN and NN: adopting the convolutional neural network (CNN) as a feature extractor in TN [7, 17, 18], compressing the linear layers of deep NN by matrix product operators [19], and implementing the convolutional operations using TN [20], etc. These attempts further motivate us to investigate the possible hybridization of TN and NN.

In this work, we incorporate the information highways (also known as shortcuts) [21, 22], non-linear activations, and dropout [23] into TN (MPS in specific), and propose Residual MPS (ResMPS in short). The essential underlying idea of ResMPS is a delicate way of inputting data such that the variational parameters of the network layers are the functions of the data features. Such idea is inspired by the traditional feed-forward neural network (FNN), while in FNN the data is input only in the initial step.

We provide two specific examples of ResMPS dubbed as simple and activated ResMPS. The simple version (sResMPS in short) is a multi-linear model that can exactly be written into a standard MPS, and the activated version (aResMPS in short) is a non-linear model equipped with NN layers. The results on fashion-MNIST show that the simple ResMPS achieves the same accuracy as MPS while its parameter complexity is half of the MPS. For the activated ResMPS, we find that the efficiency and accuracy can be significantly enhanced by introducing the non-linear activations and the dropout layers on the residual terms.

Furthermore, we determine the model interpretability of sResMPS by polynomial expression. The truncated model achieves a high level of accuracy while keeps only a few low-order terms of sResMPS. Surprisingly, the factorization [24] and exponential machines [25] have naturally emerged in this expansion scheme. ResMPS shows the underlying connections between TN and NN for ML, and can shed light on novel possibilities and flexibility of developing powerful ML models beyond NN or TN.

---

*Email addresses:* `gaochao@zjnu.edu.cn` (Chao Gao), `sjran@cnu.edu.cn` (Shi-Ju Ran)

arXiv:2012.11841v2 [cs.LG] 3 Dec 2021

## 2. Residual matrix product state

### 2.1. Definition of residual matrix product state

The traditional FNN, including the residual neural network, consists of multiple trainable layers [26]. For instance, in supervised learning, FNN maps the input sample $\mathbf{x}$ to the output $l$, e.g., sample classification. The typical form of one layer can be written as

$$\mathbf{h}^{[n]} = \sigma \left( F^{[n]} \left( \mathbf{h}^{[n-1]}; \mathbf{W}^{[n]} \right) + \mathbf{b}^{[n]} \right), \qquad (1)$$

where $\mathbf{h}^{[n-1]}$ denotes the hidden variables that are input to the $n$-th layer with $\mathbf{h}^{[0]} = \mathbf{x}$, $F^{[n]}$ denotes the mapping of the $n$-th layer (e.g., fully connected, convolution, or pooling layer). Each layer may consist some variational parameters $\mathbf{W}^{[n]}$ (weights) and $\mathbf{b}$ (bias). Furthermore, $\sigma$ denotes the activation function.

Inspired by the matrix product state [27, 28] and residual neural network [21, 22], here we propose a novel machine learning architecture dubbed as residual matrix product state (ResMPS). Different from FNN (see, Eq. (1)), ResMPS does not explicitly map the features with a feed-forward network. Instead it uses the features to parameterize FNN variational parameters. This enables the FNN to map the hidden features to the expected outputs (see, Fig. 1). In the ResMPS, the mapping of one layer is

$$\mathbf{h}^{[n]} = \mathbf{h}^{[n-1]} + v^{[n]} \left( \mathbf{h}^{[n-1]}; \mathbf{W}^{[n]}(x_n), \mathbf{b}^{[n]} \right), \quad (2)$$

where the weights $\mathbf{W}^{[n]}$ of the $n$-th layer are parameterized by the $n$-th feature $x_n$, $\mathbf{h}^{[0]}$ is simply initialized by ones, and $f^{[n]}$ denotes the map of the $n$-th layer. Therefore, the depth of ResMPS depends on the input size. Similar to the FNN [Eq. (1)], in this work we consider $v^{[n]}$ as

$$v^{[n]} = \sigma \left( L^{[n]} \left( \mathbf{h}^{[n-1]}; \mathbf{W}^{[n]}(x_n) \right) + \mathbf{b}^{[n]} \right), \qquad (3)$$

where $L^{[n]}$ is a linear map, and $\sigma$ is the activation. Similar to ResNet, the output of one layer is the addition of the output of $v^{[n]}$, and the input includes the hidden features. This is to form a shortcut of the information flow avoid the vanishing/explosion of the gradients. We further note that one obtains a standard FNN is obtained by adopting $\mathbf{h}^{[0]} = \mathbf{x}$ and removing the dependence of $\mathbf{W}$ on $\mathbf{x}$.

### 2.2. The working mechanism of ResMPS

We illustrate the path of the hidden state $h^{[i]}$ of ResMPS in the high-dimensional vector space (as shown in Fig.2a). Each layer of the ResMPS updates the state $h^{[i]}$ once to make it one step forward with shift vector $v^{[i+1]} = h^{[i+1]} - h^{[i]}$. After passes through all layers, all shift vectors are connected into a continuous path, namely $\sum_{i=1}^{N} v^{[i]}$. For the same ResMPS, different features of the samples share the same initial point (i.e., $h^{[0]}$). Since the parameter $W$ of shift vector $v$ is a function of feature $x$, the path encodes the information of samples. Besides, Similar samples have close paths in the vector space (as

shown in Fig.2b). After training convergence, samples of the same category will eventually gather together.

In order to show the consistent behavior of the path endpoint in the high-dimensional space, we use the Fashion MNIST dataset to train aResMPS, and use the tSNE algorithm [29, 30] to embed the endpoints of the ResMPS to a two-dimensional plane after the network converges. Note that before we use tSNE for dimensionality reduction, the original virtual feature has 100 components. Fig.2c illustrates the visualization of the endpoints in the two-dimensional space. It can be seen that the samples with better classification accuracy are relatively separated, while the samples with poor classification accuracy overlap with other classifications.

### 2.3. The Architecture of ResMPS

In the following, we examine two instances of ResMPS, called simple ResMPS (sResMPS, see Fig.1c) and activated ResMPS (aResMPS, see Fig.1f). The sResMPS is a multilinear model that is equivalent to MPS. It achieves the same accuracy with only half of the parameter complexity of the MPS. The aResMPS is a generalized version of sResMPS, in which the generalization efficiency is enhanced by introducing non-linear activation functions and dropout in the FNN part. The map of one layer in the sResMPS is written as

$$h_j^{[n]} = h_j^{[n-1]} + \sum_i x_n W_{ij}^{[n]} h_i^{[n-1]}. \qquad (4)$$

The weights of the layers in the FNN are linearly dependent on the features $\mathbf{x}$. The bias terms are also disabled in this example.

sResMPS is equivalent to a restricted version of MPS, which can achieve identical performance with only half parameter complexity of standard MPS. See Sec. 2.4 for details.

It is seen that MPS has a remarkable representation power. The training error is less than $1\%$ [31]. However, the gap between the training and testing accuracy suggests over-fitting issue. To address the over-fitting issue, we propose the activated ResMPS (aResMPS) by incorporating the non-linear activation functions and dropout. This also enhances the generalization power [32]. The map of each layer in the FNN of the aResMPS is more-or-less a fully-connected layer with a shortcut, which reads

$$\mathbf{h}^{[n]} = \mathbf{h}^{[n-1]} + \sigma \left( L^{[n]}(\mathbf{h}^{[n-1]}) + \mathbf{b}^{[n]} \right), \qquad (5)$$

where $\sigma$ is an activation function. The map $L^{[n]}$ rely on the feature $x_n$ in a non-linear fashion

$$L^{[n]}(\mathbf{h}^{[n-1]})_j = \sum_{c=1,2} \left[ \xi^{[c]}(x_n) \sum_i W_{ij}^{[n,c]} h_i^{[n-1]} \right], \quad (6)$$

with $\xi^{[1]}(x_n) = x_n$ and $\xi^{[2]}(x_n) = 1 - x_n$.

The architecture of ResMPS is flexible, due to the choice of $\xi^{[c]}(x_n)$ and the number of channels $\dim(c)$. We introduce $\xi^{[c]}$ to enhance the non-linearity of the aResMPS. It is worth mentioning that even sResMPS represents a non-linear map on the features $\mathbf{x}$ (but a linear map on the hidden features).

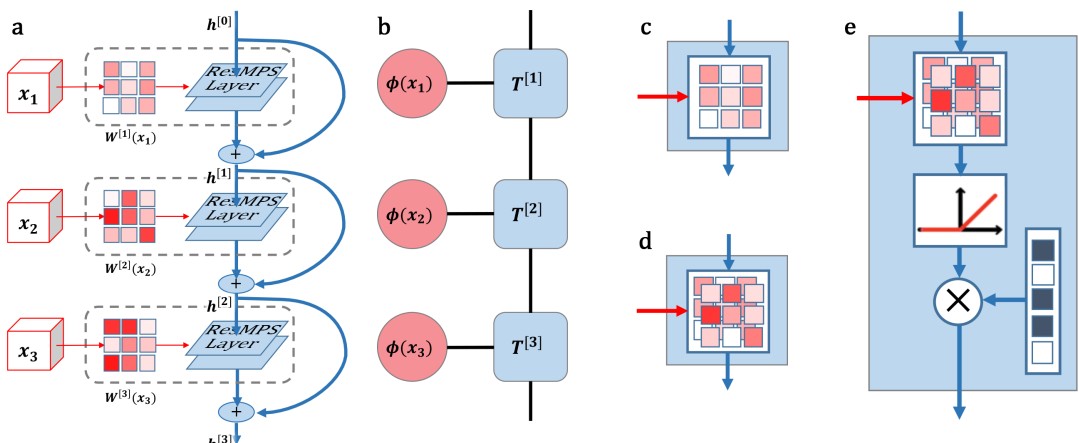

Figure 1: Illustrations of a typical ResMPS compared with a standard MPS. (a) An illustration of ResMPS containing a three-layer FNN in which the variational parameters are functions of the features, **x**. (b) An illustration of a three-tensor MPS, which is contracted with the feature vectors (see Eq. (10)). (c) An illustration of sResMPS, which is only parameterized by a single channel weight matrix. (d) An illustration of ResMPS, which is equivalent to the standard MPS. (e) An illustration of aResMPS, where the hidden feature will pass through a two-channel linear layer, ReLU activation, and dropout layer in sequence.

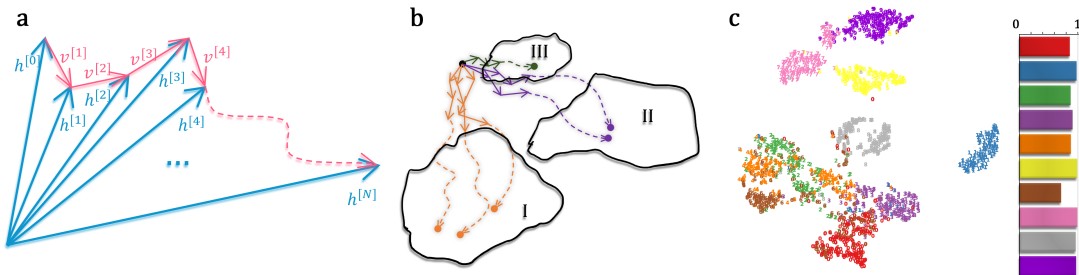

Figure 2: Encoding process of the ResMPS. (a) An illustration of a high-dimensional path of one sample. Blue arrows represent hidden features between different layers. Red arrows represent shift vectors contributed by the residual part. (b) An illustration of the aggregation behavior of samples. The same color denotes samples belong to the same class. (c) The two-dimensional data distribution generated by tSNE on the endpoint dimension reduction of (b), the data points come from the Fashion MNIST data set, and the corresponding accuracy is on the right. Note we reduce to two features in tSNE for illustration. The number of features in the actual classifications equals the number of hidden features (far larger than 2), which leads to better separations of the samples from different classes than what is visualized here.

For the aResMPS, the map on either the features or the hidden features is non-linear. Indeed, the FNN embedded inside the aResMPS is replaced by any NN. Here, we choose a standard fully-connected network with two channels labeled by $c$.

Throughout this paper, we choose the ReLU activation function that screen the negative inputs [33, 34]. Due to of its piecewise linear characteristics, the gradient directly passes through it without any attenuation or enhancement. Therefore, the ReLU function is suitable for enhancing non-linearity of the deep networks through improving its expression ability and avoiding vanishing/explosion of the gradient. Furthermore, we use dropout combining with the residual structure to improve the generalization ability of ResMPS. This is to create an ensemble of networks, while avoiding the co-adaptation of intermediate variables [23, 35, 36]. We impose dropout on the residual terms, i.e. $\mathbf{h}^{[n]} = \mathbf{h}^{[n-1]} + \text{dropout}\left(\sigma\left(\cdots\right)\right)$.

If we discard the activation and the dropout layers of aResMPS (see Fig.1e), we will get a standard two-channel MPS. For a standard MPS with physical bond dimension $d = 2$, the map given by a local-thensor constraction is [31]

$$h_j^{[n]} = \sum_{c=1,2}\left[\xi^{[c]}(x_n)\sum_i T_{ij}^{[n,c]}h_i^{[n-1]}\right]. \tag{7}$$

If we introduce transformation $T_{ij}^{[n,c]} = W_{ij}^{[n,c]} + \delta_{ij}$, we can simply get $h_j^{[n]} = h_j^{[n-1]}\left(\sum_{c=1,2}\xi^{[c]}(x_n)\right) + \sum_{c=1,2}\left[\xi^{[c]}(x_n)\sum_i W_{ij}^{[n,c]}h_i^{[n-1]}\right]$. Take feature map with norm-1 normalization [31], i.e. $\sum_{c=1,2}\xi^{[c]}(x_n) = 1$, we get a ResMPS with map

$$h_j^{[n]} = h_j^{[n-1]} + \sum_{c=1,2}\left[\xi^{[c]}(x_n)\sum_i W_{ij}^{[n,c]}h_i^{[n-1]}\right]. \tag{8}$$

### 2.4. Benchmarking results

For the MNIST [39] and fashion-MNIST [40] datasets, Table 1 shows the accuracy of the sResMPS and aResMPS, compared with several established NN [38] and TN models [7, 9, 15, 17, 31, 37]. As it is seen the MPS and ResMPS models represent high level of representation as indicated by their high training accuracy. The aResMPS also surpasses the probabilistically interpretable Bayesian [15] and other TN models, including the two-dimensional TN known as projected-entangled pair state (PEPS) [17]. It also achieves a (slightly) better accuracy than that of CNN-PEPS model, in which CNN is adopted as the feature extractor. This accuracy surpasses the CNN without the stacking architecture, such as AlexNet [38]. The aResMPS still does not overperform the ResNet which is formed by stacking multiple convolution layers. It seems that the ResMPS models eventually surpass ResNet by replacing the fully-connected network with more sophisticated ones or staking multiple ResMPSs.

To see the equivalence to the standard MPS and sResMPS mentioned in Sec.2.3, let us introduce the third-order tensors

$\mathbf{T}^{[n]}$ satisfying

$$T_{1,:,:}^{[n]} = \mathbf{I}, \quad T_{2,:,:}^{[n]} = \mathbf{W}^{[n]}. \tag{9}$$

The feature vectors $\phi(x_n)$ are obtained by the feature map as $\phi(x_n) = (1, x_n)$, similar to Refs. [5, 9, 31]. Therefore, the sResMPS is equivalent to the standard MPS formed by the following tensors

$$\mathcal{T} = \sum_{\{a\}}\prod_n T_{p_n a_n a_{n+1}}^{[n]} \tag{10}$$

as its tensor-train cores [41] [Fig. 1 (b)]. The numbers of the input and output hidden features for different layers provide the two virtual bond dimensions of the MPS, i.e., $\{\dim(a_n)\}$. In this work, we fix $\dim(a_n) = \chi, \forall n$. The physical dimension of the MPS should also match the dimension of the feature vector, i.e. $\dim(\phi(x_n)) = \dim(p_n)$.

For $\dim(p_n) = 2$, the number of variational parameters in sResMPS is $\sim O(N\chi^2)$ where $N$ is total number of features. This is only half of that in the MPS which is $\sim O(2N\chi^2)$. Our numerical simulations show that the accuracy of both models is almost the same. See the training and testing accuracy versus epochs on fashion-MNIST dataset [40] in Fig. 3 (a) with $\chi = 40$. This is because one of the two channels of each tensor in the MPS is much less "activated". The inset of Fig. 3 (a) shows the average norm of the two channels of different tensors

$$q_p^{[n]} = \frac{1}{\chi^2}\sum_{j=1}^{\chi}\sum_{k=1}^{\chi}\left|T_{pjk}^{[n]} - \delta_{jk}\right|, \tag{11}$$

with $p = 1, 2$ representing the channels. The main contribution to the output is from the second channel. Therefore, one channel is sufficient to propagate the information to the output.

In physics, the virtual bond dimension, $\chi$, characterizes the representation power of the MPS. This is because it determines the total number of variational parameters and the upper bound of the entanglement entropy the MPS can carry [1]. This may not be the case for machine learning. We show this by adding masks on the variational parameters, i.e., pruning [14]. Each parameter is multiplied by a factor that is either zero or one. The parameters multiplied by zeros are masked. To mask a certain number of parameters, we choose to mask those with relatively small absolute values. We then optimize the unmasked parameters after the masks taking effect.

Fig. 3 (b) shows the accuracy values versus the number of unmasked parameters $M$. For different virtual bond dimensions, $\chi = 20, 30$, and 40, the results are similar if the number of the unmasked parameters are the same. This suggests that the parameter which characterizes the representation and generalization power, is in fact, M (not $\chi$). For a the given $\chi$, it is possible to further reduce the complexity of MPS (and sResMPS) without harming the accuracy. Our results also indicate that the sResMPS achieves its maximal representation power for $M \sim O(10^4)$ (the training accuracy $\simeq 99.98\%$).

Table 1: Experimental results on MNIST and Fashion-MNIST dataset. The first 6 models are prune TN architectures, while AlexNet, ResNet, and CNN-PEPS are NN or TN-NN hybrid models. For aResMPS, we use RELU as activation function.

| Model | MMIST train | MMIST test | Fashion-MMIST train | Fashion-MMIST test |
|---|---|---|---|---|
| MPS machine [31] | 1.0000 | 0.9855 | 0.99 | 0.88 |
| Unitary tree TN [9] | 0.98 | 0.95 | - | - |
| Tree curtain model [37] | - | - | 0.9538 | 0.8897 |
| Bayesian TN [15] | - | - | 0.8950 | 0.8692 |
| EPS-SBS [7] | - | 0.9885 | - | 0.886 |
| PEPS [17] | - | - | - | 0.883 |
| CNN-PEPS [17] | - | - | - | 0.912 |
| AlexNet [38] | - | - | - | 0.8882 |
| ResNet [38] | - | - | - | 0.9339 |
| sResMPS(+dropout) | 1.0000 | 0.9898 | 0.9920 | 0.9076 |
| aResMPS(+ReLU,+dropout) | 1.0000 | 0.9900 | 0.9999 | 0.9146 |

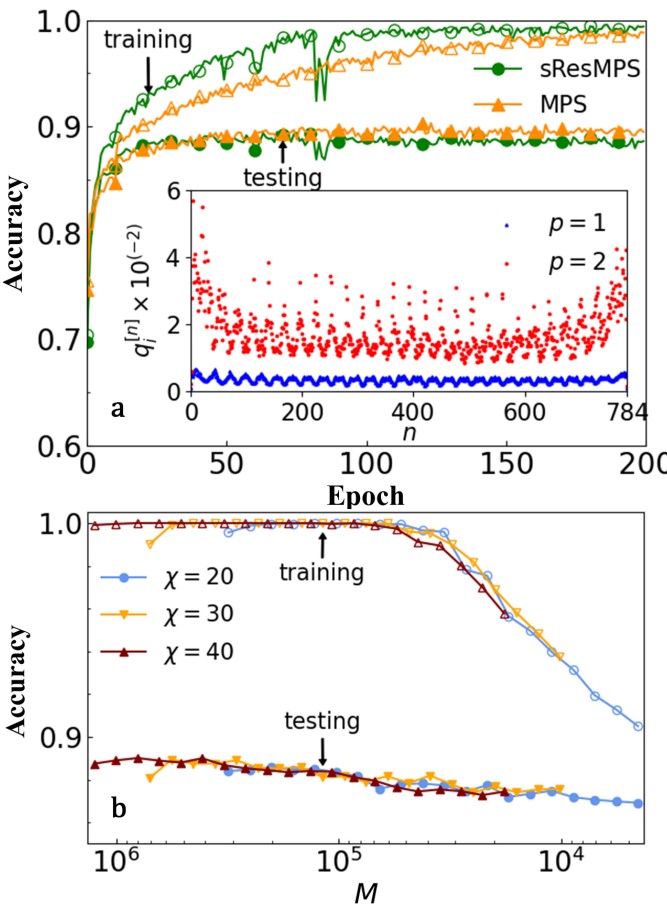

Figure 3: Numerical results of the simple ResMPS. (a) Training and testing accuracy of sResMPS (without dropout) and MPS versus epochs on the fashion-MNIST dataset. The inset shows the average norm [Eq. (11)] of the two channels in the MPS for different tensors $n$. (b) Training and testing accuracy of the sResMPS versus the total number of the unmasked weights in the sResMPS. The left end of each curve corresponds to the un-pruned result. It is also seen that the first few steps of pruning improve the accuracy. Note the total number of sResMPS with $\chi = 20$, 30, and 40 equals to about $3 \times 10^5$, $7 \times 10^5$ and $13 \times 10^5$, respectively.

## 3. Properties of the residual structure

### 3.1. Avoiding the gradient problems by residual terms

A typical MPS architecture that is designed for pattern recognition contains hundreds of tensor cores. Such an architecture probably encounters the gradient vanishing/exploding problems. For this reason, some existing MPS schemes apply a DMRG-like algorithm where the MPS takes the canonical form [5, 6, 10, 42]. In these attempts, however, the accuracy is sensitive to the hidden features' dimensions (virtual bonds). Recently, an MPS algorithm was proposed based on automatic gradient technique [31] that can achieve higher accuracy than that of the previous ones, while its performance is not sensitive to the virtual dimensions. To find the reason that such a deep network avoids the gradient problems, here we construct the tensor cores to satisfy a special form given by Eq. (9). The identity in $T_{1,:,:}^{[n]}$ plays the role of "highway" to pass the information from the previous tensor core directly to the latter ones. The components $T_{2,:,:}^{[n]}$ represent the residual terms, which is $\ll O(1)$. The application of residual condition implies that each layer of ResMPS can easily express identity mapping. In other words, the architecture of ResMPS satisfies the identity parameterization [21, 22, 43].

To further demonstrate the role of identity parameterization in ResMPS, we use Gaussian distributions with zero mean and standard deviation $\varepsilon$ to randomly initialize the elements of $T_{2,:,:}^{[n]}$. Fig. 4 shows the testing accuracy at the 10-th, 20-th and 50-th epochs. For a sufficiently small $\varepsilon$, the accuracy is quickly and stably converged. However, for relatively large $\varepsilon$ (e.g., $O(10^{-1})$) which is illustrated by the red region, the gradients become unstable. Consequently, the accuracy stays around 0.1 and cannot be further improved by the training process. Not that this may be unstable in most cases if instead of the identity parameterization, the entire $T$ is randomly initialized.

### 3.2. Relations to polynomial expansion

The forward propagation of the sResMPS (4) is fully linear on the hidden features. Applying the maps to the initial hidden features $\mathbf{h}_0$ in sequence, we can then rewrite the output hidden

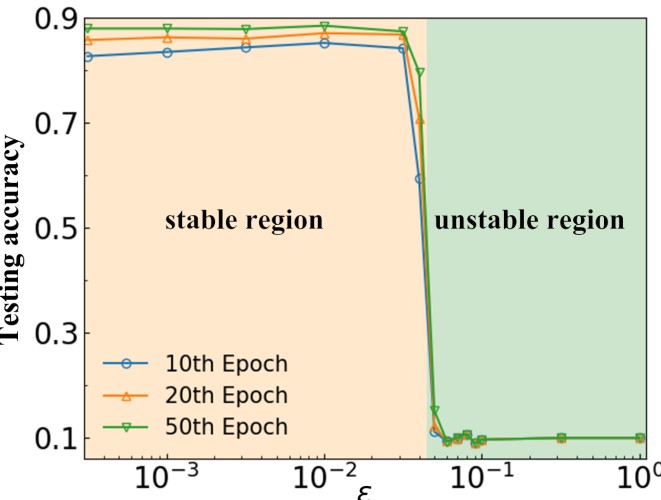

Figure 4: The testing accuracy of the sResMPS versus $\varepsilon$ on fashion-MNIST dataset. Here $\varepsilon$ is the standard deviation of the initial residual part. We fix the number of epochs to be 10, 20, and 50. The network can be trained stably for small values of $\varepsilon$. Otherwise, the training process encounters gradient vanishing (or explosion) problems. The stable and unstable regions are illustrated by green and red colors, respectively. Note that for the red region, the value of the network elements is diverged, The network will give classifications randomly, thus, the accuracy tends to 0.1.

features in an expansive form [Fig. 5 (b)] as

$$\mathbf{h}^{[N]} = \left(\mathbf{I} + x_N \mathbf{W}^{[N]}\right) \dots \left(\mathbf{I} + x_2 \mathbf{W}^{[2]}\right) \left(\mathbf{I} + x_1 \mathbf{W}^{[1]}\right) \mathbf{h}^{[0]}$$
$$= \sum_{k=0}^{N} \mathbf{M}^{[k]} \mathbf{h}^{[0]}, \quad (12)$$

where $N$ is the total number of features $\mathbf{x}$. The output $\mathbf{h}^{[N]}$ is the stack of $N$ terms. The zeroth term satisfies $\mathbf{M}^{[0]} = \mathbf{I}$, which is the result of the information highway from the first input hidden features to the output. The term $\mathbf{M}_1 = \sum_{\alpha=1}^{N} x_\alpha W^{[\alpha]}$ is the part in ResMPS which is linear on the features $\mathbf{x}$. The $k$-th term contains the $k$-th order contributions from $\mathbf{x}$, i.e.,

$$\mathbf{M}_k = \sum_{\alpha_1 \dots \alpha_k = 1}^{N} G_{\alpha_1 \dots \alpha_k} x_{\alpha_1} \dots x_{\alpha_k} \mathbf{W}^{[\alpha_1]} \dots \mathbf{W}^{[\alpha_k]} \quad (13)$$

$$G_{a_1, a_2, \dots, a_n} = \begin{cases} 1, & a_1 > a_2 > \dots > a_n \\ 0, & \text{otherwise.} \end{cases} \quad (14)$$

This formula is a specific form of the Exponential Machines [25]. Due to their essential similarity, the algebraic properties of Exponential Machines are also valid for sResMPS. For instance, the output feature $\mathbf{h}^{[N]}$ is a linear mapping concerning the initial hidden feature $h_0$, and a multi-linear mapping concerning the feature $\mathbf{x}$.

From the residual condition (see Eq. (9) with $|\mathbf{W}^{[n]}| \ll O(10^{-1})$), the contributions from the higher-order terms of (13) should decay exponentially with $k$. Therefore, we can define a set of lower-order effective models by retaining the first few terms. For instance by only keeping the zeroth- and first-order terms in Eq. (13), we simply obtain a model in which the output

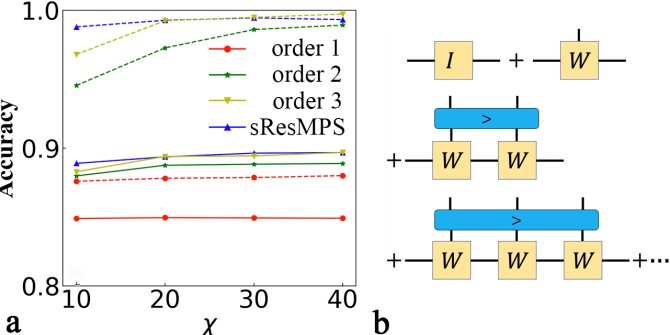

Figure 5: Polynomial expansion based on ResMPS. (a) Training and testing accuracy versus $\chi$ by taking different orders in the expansion form; (b) The illustration of the polynomial expansion picture of the sResMPS. See Eq. (13).

features are linear to both hidden and sample features. Keeping the zeroth, linear, and quadratic terms the resulting model is

$$\mathbf{h}^{[N](2)} = \left(\mathbf{I} + \sum_{\alpha=1}^{N} x_\alpha \mathbf{W}^{[\alpha]} + \sum_{\alpha,\beta=1}^{N} G_{\alpha,\beta} x_\alpha x_\beta \mathbf{W}^{[\alpha]} \mathbf{W}^{[\beta]}\right) \mathbf{h}^{[0]}.$$
$$(15)$$

This model is similar to Factorization Machines [24] and polynomial NN [44].

Fig. 5 (a) shows the difference between the accuracy of several lower-order models and the sResMPS. This implies that the significant improvement achieved by the sResMPS has its root in a few lower-order terms, especially the linear term. As the order increases, the cost of directly computing Eq. (12) is also exponentially increased. Therefore, truncating the order of expansion is not economical. ResMPS adopts a different and efficient scheme for retaining all higher-order interactions.

## 4. Conclusion

We propose ResMPS by incorporating MPS with the information highways, non-linear activations, and dropout. In contrast to from FNN, the variational parameters in ResMPS are replaced by adjustable functions. For FNN, features are input at the first layer of the network. For ResMPS, however, features are divided and input into the weight matrices of each layer, which is inherited from MPS. Furthermore, the introduction of the neural network structures results in ResMPS to have a more vital expression ability than the MPS. We also present two specific versions of ResMPS.

The first derived architecture sResMPS, is a simple linear version of ResMPS. By comparing MPS' learning performance on the fashion-MNIST dataset, we further reveal the channel redundancy of MPS. sResMPS also discards the redundant channel. Consequently, it achieves consistent accuracy while the parameter complexity is halved.

The second one is aResMPS, which is the general ResMPS equipped with activation and dropout layers. We further compare the model with several TN and NN models on the fashion-MNIST dataset. The activation and dropout layer enhance the non-linearity and generalization ability of the model, respectively. Therefore, aResMPS surpass the state-of-the-art TN

methods and AlexNet in terms of accuracy, although still inferior to ResNet that is formed by stacking multiple convolution layers. Going beyond present aResMPS to achieve higher accuracy, e.g. replacing the weight matrices with convolution layers, is a valuable improvement direction of ResMPS.

The perspectives of the residual network derived the polynomial expansion of ResMPS. The benefits are two-fold. Firstly, we give the condition of vanishing/explosion of the gradients of ResMPS. This helps the feature design of MPS and ResMPS algorithms with stable convergence. Secondly, it establishes the equivalence between MPS and polynomial networks such as Factorization Machines and Exponential Machines. Further numerical evidence suggests that the contribution of high-order terms is insignificant. This helps to better understand the MPS and ResMPS.

Are other NN structures (e.g., convolution and pooling layers) compatible with ResMPS? Is it possible to propose a ResMPS structure based on general NN structures (e.g., Tree TN or Projected Entangled-Pair States)? These problems are worthy of further investigation in the future.

## 5. Acknowledgements

Y.-M.M. and C.G. are supported by National Natural Science Foundation of China (NSFC, Grant No. 1183501 and No. 12074342) and Zhejiang Provincial Natural Science Foundation of China (Grant No. LY21A040004). S.-J.R. is supported by NSFC (Grant No. 12004266 and No. 11834014), Beijing Natural Science Foundation (No. 1192005 and No. Z180013), Foundation of Beijing Education Committees (No. KM202010028013), and the Academy for Multidisciplinary Studies, Capital Normal University. J.Z. and P.Z. are supported by NSFC (Grant No. 61772363).

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
