# Peer review of "Residual Matrix Product State for Machine Learning"

_SciPost Physics_

## Round 2 · Referee Report · Anonymous (Referee 1) · 2022-1-7

Strengths

  1. The paper gives an empirical study of the impact of several architecture changes in matrix product state (MPS) models, a special form of tensor network (TN), on their performance in image classification tasks. Given the newness of TN models in machine learning, these types of empirical investigations are important for furthering the adoption of these physics-inspired models within that community.

  2. The use of residual connections has succeeded in delivering better performance in deep neural networks (NNs), and the authors give evidence that such connections are also beneficial in the context of MPS.

  3. The empirical results give a comparison to a wide range of previous TN and NN model baselines.

  4. The empirical results also include several interesting analyses involving masking MPS parameters, truncating the model's output via a polynomial expansion, and varying the initialization noise. Although not essential to the paper's broader message, these results are useful for practitioners looking to most effectively use these models for solving real-world learning problems.

Weaknesses

1a. The paper tries to distinguish three special classes of MPS, namely "standard" MPS (MPS), "simple" residual MPS (sResMPS), and "activated" residual MPS (aResMPS), which are experimentally compared in supervised learning on the MNIST and FashionMNIST datasets. My biggest complaint about the paper is the poor/inconsistent demarcation between these models, and how they are compared in the main experimental results. This significantly weakens the main claim of the paper, that residual layers in MPS give better performance. The quick summary of this criticism (which I elaborate on below, and in the requested changes section) is that the paper's experimental methodology at present makes it impossible to determine how much of the reported performance gains are due to the use of residual connections vs. dropout layers vs. nonlinear activation functions vs. embedded inputs (i.e. mapping a scalar feature of input data into a vector of dimension d>1).

1b. It appears that the intended division between these three architectures is that sResMPS = MPS + residual connections - embedded inputs, while aResMPS = MPS + residual connections + dropout + nonlinear activations. This is already problematic, as it doesn't permit the influence of residual connections vs. embedded inputs (for example, using the simple d=2 embedding of (1, x) appearing below Equation 9) to be assessed. However, while some experiments (Figure 3) use this version of sResMPS, the main experiments (Table 1) use a different version, where dropout is added to the sResMPS model. While the results of Figure 3 show sResMPS performing similarly to standard MPS (this is the origin of the claim that ResMPS give similar performance to standard MPS with only half the parameters), the more comprehensive results in Table 1 are ambiguous here, because of this additional use of dropout.

1c. The use of ReLU activations (which forfeit the model's claim of being a real TN) in aResMPS appear to boost the performance of the model, but the evidence for this isn't solid. In particular, the difference between the last two rows of Table 1 arises from the combination of using ReLUs and using embedded inputs.

  1. Many important experimental details are missing, such as the type of optimization used (presumably stochastic gradient descent), the language and library used for training (e.g. TensorFlow in Python), learning rates, the choice of optimizer, dropout probabilities, regularization (if any), image preprocessing, etc. More importantly, the code for the paper's experiments isn't available anywhere, preventing interested readers from reproducing these results.

Report

The paper describes a breakthrough for enabling better performance within matrix product state models when applied to classical data. This poor performance is something that is currently a major stumbling block in the adoption of tensor network models within machine learning, so finding simple techniques that already give comparable performance to state-of-the-art deep learning models from ~5 years ago is significant.

On the other hand, the paper currently doesn't meet acceptance criterion 5, as its experimental results aren't reproducible. Consequently, the authors should include all necessary experimental details (including their code) in order to fully meet the journal's acceptance criteria.

Requested changes

I am ordering my changes by rough order of importance, with more important changes being given first.

  1. Add an appendix or additional (sub)section containing all of the experimental details needed to reproduce these results (see my comment in the Weaknesses section for a few details to include). If there aren't any issues with proprietary/confidential source code, then please make the experimental code available to readers through a link to a Github repository or some other equivalent means.

  2. It isn't clear if the results in Table 1 for the "MPS machine" (i.e. standard MPS) is from an implementation by the authors, or is reported from [31] (Efthymiou et al. 2019). In the latter case, I would request that the authors implement this baseline in their own code, in order to mitigate the impact of any differences in experimental setup between [31] and the present work. Given that both ResMPS models the authors consider are shown to reduce to standard MPS in a certain configuration, this shouldn't be hard to do.

  3. Similarly, the performance of (1) sResMPS without dropout, and (2) aResMPS without ReLU activations should be reported in Table 1. Adding these additional models will give a type of ablation study, which is crucial for understanding how dropout, input embedding, and residual layers individually impact the performance of MPS models.

  4. In the caption to Figure 4 it is stated "The stable and unstable regions are illustrated by green and red colors, respectively". However, the figure uses orange for the stable region, and green for the unstable region. This should be fixed.

  5. In the caption to Table 1, it is stated "The first 6 models are prune TN architectures". It isn't clear what this means, and a straightforward reading from the text would imply that the parameters of these models have been masked using the pruning procedure reported in Figure 3b. I'm not confident this is what the authors mean though, so they should clarify/fix this sentence.

  6. In Equation 2, $v^{[n]}$ is used to indicate the function mapping the $n-1$'th hidden state and layer weights to the residual component of the $n$'th hidden state, but this function is described below the equation as being $f^{[n]}$. In Equation 3, $v^{[n]}$ is then used to indicate the vector representing this non-residual component. This usage is inconsistent, and the authors should fix this.

  7. In Figure 5, the fact that training (test) accuracies are reported with dotted (solid) lines should be described in the caption.

  8. In the caption to Figure 3, I believe the text "Note the total number of sResMPS" should be "Note the total number of parameters of sResMPS".

  9. Change the four uses of "MMIST" in Table 1 to "MNIST".

  10. In the text above Equation 7, "thensor" should be "tensor".

  • validity: good
  • significance: good
  • originality: good
  • clarity: good
  • formatting: excellent
  • grammar: reasonable

Author:  Yeming Meng  on 2022-09-09  [id 2798]

(in reply to Report 1 on 2022-01-07)
Category:
answer to question
correction

MAJOR CHANGES

  1. Lines 76-81. We added a description of the structure of the whole paper.
  2. Lines 97-101. We rewrote these sentences to make them clearer.
  3. Lines 107-111. We added descriptions of an output linear layer that maps the final hidden $\hat{h}^{[N]}$ to classifications.
  4. Lines 127-131 & Fig.3. We added visualization and descriptions of t-SNE intermediate data.
  5. Lines 189-191. We added more formulas to state the equivalence of sResMPS and regular MPS.
  6. Lines 222-242. We added Sec. 2.5 to compare ResMPS, RNN, and transformer network.
  7. Appendix A. Detailed training information which helps to reproduce our result.
  8. We uploaded codes to GitHub repository (see YemingMeng/ResMPS).
  9. Appendix B. Comparison of different paths which map 2D figures to 1D sequence.
  10. Appendix C & Tab. 2. Additional benchmarks 1 that compare exhaustive combinations of different components (channels, activation and dropout).
  11. Appendix C. Additional benchmark 2 that compare performance under small virtual feature dimension $\chi<10$.
  12. Several typos were fixed.

STRENGTHS

We thank the referee for pointing out the strengths of our manuscript. The summary of strengths is comprehensive and accurate. By the way, the arXiv version of our article has already drawn some attention in the physics community and has been cited by some papers published in physics journals, including Frontiers in Physics (https://www.frontiersin.org/articles/10.3389/fphy.2022.885402/full), Quantum Machine Intelligence (https://link.springer.com/article/10.1007/s42484-022-00081-1), and one uploaded in arxiv (https://arxiv.org/abs/2208.06029).

WEAKNESSES

We thank the referee for pointing out these weaknesses. These concern a gap between the construction of sResMPS and aResMPS, and the description of the training details for reproducing our results. We took care of these issues in the updated version by adding more benchmarks, which fill the gap between sResMPS and aResMPS with benchmark data (major change 10), and supplementing the missing training details (major changes 3&7). We also upload our codes at GitHub repository (major changes 8, YemingMeng/ResMPS). After adding these details, we believe the current version is much more friendly to the readers who intend to reproduce our results.

Point 1) The original experiment is indeed for demonstration and more models should be included. We fixed this issue by adding more benchmarks, please refer to the major change 10.

Point 2) We also fixed this issue, please refer to major changes 7&8.

REPORT

We thank the referee for bringing up these issues. Furthermore, we thank the referee for pointing out the scientific importance of our work. We have performed proper revisions and extensions based on the valuable comments and suggestions from both referees.

REQUESTED CHANGES

  1. Add an appendix or additional (sub)section containing all of the experimental details needed to reproduce these results (see my comment in the Weaknesses section for a few details to include). If there aren't any issues with proprietary/confidential source code, then please make the experimental code available to readers through a link to a Github repository or some other equivalent means.

Reply: We added appendix A titled "Training details" to give more details, and our codes are now available on GitHub repository YemingMeng/ResMPS. Please refer to the major changes 7&8.

  1. It isn't clear if the results in Table 1 for the "MPS machine" (i.e. standard MPS) is from an implementation by the authors, or is reported from [31] (Efthymiou et al. 2019). In the latter case, I would request that the authors implement this baseline in their own code, in order to mitigate the impact of any differences in experimental setup between [31] and the present work. Given that both ResMPS models the authors consider are shown to reduce to standard MPS in a certain configuration, this shouldn't be hard to do.

Reply: We wrote our own codes to realize the MPS machine in Ref. [31]. We found that by taking our hyper parameter and feature map, the accuracy is improved by about 2% compared with the reported results in Ref. [31].

  1. Similarly, the performance of (1) sResMPS without dropout, and (2) aResMPS without ReLU activations should be reported in Table 1. Adding these additional models will give a type of ablation study, which is crucial for understanding how dropout, input embedding, and residual layers individually impact the performance of MPS models.

Reply: We added a new appendix/section to include more benchmark details. In simple words, both the ReLU activation and dropout can improve the accuracy of ResMPS, and their combination achieves the best performance. Please refer to the major change 10.

  1. In the caption to Figure 4 it is stated "The stable and unstable regions are illustrated by green and red colors, respectively". However, the figure uses orange for the stable region, and green for the unstable region. This should be fixed.

Reply: We fixed this issue in the new version. These two regions are now correctly referred to.

  1. In the caption to Table 1, it is stated "The first 6 models are prune TN architectures". It isn't clear what this means, and a straightforward reading from the text would imply that the parameters of these models have been masked using the pruning procedure reported in Figure 3b. I'm not confident this is what the authors mean though, so they should clarify/fix this sentence.

Reply: We fixed this issue in the new version. We added several descriptions, which are "The first 6 models are pure TN architectures, which means they are multi-linear, and no neural structures like pooling, activation and convolutional are introduced".

  1. In Equation 2, $v^{[n]}$ is used to indicate the function mapping the $n−1$'th hidden state and layer weights to the residual component of the $n$'th hidden state, but this function is described below the equation as being $f^{[n]}$. In Equation 3, $v^{[n]}$ is then used to indicate the vector representing this non-residual component. This usage is inconsistent, and the authors should fix this.

Reply: We apologize for this ambiguity. We intended to say by $\mathbf{v}^{[n]}(\mathbf{h}^{[n]}; \mathbf{W}^{[n]}, \mathbf{b}^{[n]})$ in Eq. (2) that $\mathbf{v}^{[n]}$ is the function of $\mathbf{h}^{[n]}$ parametrized by $\mathbf{W}^{[n]}$ and $\mathbf{b}^{[n]}$. Eq. (3) gives a concrete function. We added several words below Eq. (2) to give a better explanation on these equations.

  1. In Figure 5, the fact that training (test) accuracies are reported with dotted (solid) lines should be described in the caption.

Reply: We added the missing description to the caption. The new caption is now "Training (dashed) and testing (solid) accuracy versus ...".

  1. In the caption to Figure 3, I believe the text "Note the total number of sResMPS" should be "Note the total number of parameters of sResMPS".

Reply: We have adopted a new description.

  1. Change the four uses of "MMIST" in Table 1 to "MNIST".

Reply: We fixed this typo.

  1. In the text above Equation 7, "thensor" should be "tensor".

Reply: We fixed this typo.

---

## Round 2 · Referee Report · Anonymous (Referee 2) · 2022-2-24

Strengths

1- The authors introduce two new deep learning architectures, sResMPS and aResMPS, as a generalization of the MPS-inspired models that are being explored as alternative to neural networks.

2- The authors demonstrate that their models achieve state-of-the-art performance on the benchmark task of classifying Fashion-MNIST.

3- The authors develop intuition for their ansatz by discussing limiting cases in which it becomes equivalent to MPS and connections to Exponential and Factorization Machines.

Weaknesses

1- The authors present their architectures as generalizations of MPS inspired by the ResNet architecture. However, in my view the models they construct exhibit a very close similarity with the family of RNN-like models and Transformers (https://arxiv.org/abs/1706.03762). This connection is not discussed in the manuscript.

2- In problems of quantum physics MPS are designed for applications in one-dimensional systems and it is known that their extension to two dimensions is at least very challenging. This is due to the necessary choice of an ordering of the degrees of freedom and an intrinsic limitation on the correlation length that can be encoded. Therefore, the chosen benchmark problem of image recognition does not seem to be a machine learning task, where one would naturally expect MPS-like models to excel. Including a benchmark problem with one-dimensional data could help to better illustrate the strengths and weaknesses of the approach.

3- Some details of the introduced architectures remain unclear and the manuscript lacks information about the training procedure, which makes it hard to reproduce the results.

Report

The authors introduce new deep learning architectures, which they devised as a generalization of MPS inspired by the ResNet architecture. They demonstrate better performance than what was achieved with other tensor network based models on the benchmark problem of Fashion-MNIST.

The results seem to be valid (although further information is needed for reproducibility) and the design choices in constructing these models are interesting. However, I have two main objections to the publication of this manuscript in SciPost Physics:

1- The topic of the paper is hardly related to physics. It is the proposal and empirical investigation of a new deep learning architecture as it is typically done in the machine learning community. The link to physics is a limit in which the new architecture becomes equivalent to MPS (or maybe better tensor train). But besides this formal similarity none of the physical concepts related to MPS are used in the presentation and analysis of the results. Therefore, I am doubtful whether SciPost Physics is the right platform to publish this research.

2- The presented architectures have a close similarity to RNNs and Transformer Networks. Since the authors do not discuss this at all and they do not include those when comparing performance, it is at this point difficult for me to judge the significance of the presented new architectures.

I recommend a rejection based on my point 1 above (I believe there is no suited alternative journal in the SciPost family). Otherwise, point 2 and the further requested changes should be addressed in a major revision.

Requested changes

The following questions/remarks should be considered when revising the manuscript:

1- Discuss the relation of ResMPS to RNNs and Transformer Networks. Include performance of such networks on the benchmark problem if possible.

2- Include all details about the training procedure.

3- While the intermediate transformation of hidden features is well explained, it remained unclear to me how the final output that indicates the classification of the given data is produced by the network. This should be clarified.

4- It would be helpful to initially clarify the domain of the training data $x$.

5- For the ResMPS a one-dimensional ordering of the image data has to be chosen. How is this done? It could be interesting to investigate how the performance depends on the ordering.

6- At the end of section 2.2 you claim that the path of hidden features is similar for similar images. Then, you present a t-SNE for the final hidden features. Can you include a t-SNE of intermediate hidden features to prove the claim that already the paths are similar?

7- Regarding the factor of 2 between the parameter efficiency of MPS and sResMPS: Is this because the sResMPS fixes a gauge choice, which is left free in the MPS?

8- The discussion around Eq. (9) and (10) seems to be inconsistent with the definition of sResMPS in Eq. (4). In particular, how does the feature map $\phi(x)$ enter Eq. (4)?

9- At the end of section 2.4 the expressivity of the ResMPS as function of $\chi$ and $M$ is discussed, coming to the conclusion that $\chi$ does not affect the expressivity of the ResMPS. However, only three large values of $\chi$ are considered and Fig. 3 shows that the expressivity is also not affected by varying $M$ over about one order of magnitude. How does the performance change when you vary $\chi$ between 1 and 20?

10- The text contains a non-negligible amount of typos and grammatical errors. Please proof-read it thoroughly.

11- Section 2.4 could be further divided into subsections for better clarity. In the current form, three different aspects with no immediate connection are discussed, which makes it hard to follow.

  • validity: high
  • significance: ok
  • originality: high
  • clarity: ok
  • formatting: excellent
  • grammar: acceptable

Author:  Yeming Meng  on 2022-09-09  [id 2799]

(in reply to Report 2 on 2022-02-24)

MAJOR CHANGES

  1. Lines 76-81. We added a description of the structure of the whole paper.
  2. Lines 97-101. We rewrote these sentences to make them clearer.
  3. Lines 107-111. We added descriptions of an output linear layer that maps the final hidden $\hat{h}^{[N]}$ to classifications.
  4. Lines 127-131 & Fig.3. We added visualization and descriptions of t-SNE intermediate data.
  5. Lines 189-191. We added more formulas to state the equivalence of sResMPS and regular MPS.
  6. Lines 222-242. We added Sec. 2.5 to compare ResMPS, RNN, and transformer network.
  7. Appendix A. Detailed training information which helps to reproduce our result.
  8. We uploaded codes to GitHub repository (see YemingMeng/ResMPS).
  9. Appendix B. Comparison of different paths which map 2D figures to 1D sequence.
  10. Appendix C & Tab. 2. Additional benchmarks 1 that compare exhaustive combinations of different components (channels, activation and dropout).
  11. Appendix C. Additional benchmark 2 that compare performance under small virtual feature dimension $\chi<10$.
  12. Several typos were fixed.

STRENGTHS

We thank the referee for the recognition of the state-of-the-art accuracy we have achieved, as well as the connection between tensor networks and Exponential and Factorization Machines that we revealed. Besides, we believe that the study on the gradient issue will also benefit the future developments of the TN machine learning.

WEAKNESSES

We thank the referee for pointing out these weaknesses. The main description of the ResMPS in our manuscript is given in the neural-network fashion. These would make our manuscript more friendly to the readers in the areas of AI and computer sciences. Moreover, some training details should have been provided to make the results easy to reproduce.

We for the first time showed that the some physical concepts fails in machine learning MPS, including the working machanism of matrix product state for machine learning, the failure of bound dimension $\chi$ to measure the representation ability, the advantage of residual condition to the training stability, the result of reordering test, etc. We believe that these results are heuristic and stimulative for the uses of MPS in both the physical and computational problems.

For point 1), please refer to our answer above.

For point 2), the native quantum physics MPS fails in 2D, because long range correlation exists when reordering a 2D system to a 1D chain. However, our numerical result shows that the ResMPS is not sensitive to the order of feature sequence (See appendix Path independency), which implies that correlation interpretation is invalid for ResMPS, possibly thanks to the residual structure. On the other hand, it is still an open question whether residual-based MPS can have better performance in the physical problems.

For point 3), we apologize for the missing details that should have been included in our manuscript. In the updated version these are included in appendix A.

REPORT

We thank the referee's comments on the link of our manuscript to physics. This further stimulated us to unearth more value for the physics community. We summarize the important points so-far we have found below:

1) our manuscript reveals the latent connection between MPS and feed-forward neural networks, which will in particular inspire those who work in the crossing field of tensor network for quantum physics and machine learning;

2) ResMPS under unitary condition is hopeful to be realized in some quantum platforms by, e.g., transforming into quantum circuit models.

3) The tensor network community has been devoted to bring physical concepts in machine learning, while our paper provides a new perspective on the quantum-inspired machine learning based on tensor network.

Based on the above points, we believe that this work can be interesting for both the physics community and the machine learning community. As a circumstantial evidence, the arXiv version of our article has already drawn some attention in the physics community and has been cited by some papers published in physics journals, including Frontiers in Physics (https://www.frontiersin.org/articles/10.3389/fphy.2022.885402/full), Quantum Machine Intelligence (https://link.springer.com/article/10.1007/s42484-022-00081-1), and one preprinted in arxiv (https://arxiv.org/abs/2208.06029).

REQUESTED CHANGES

  1. Discuss the relation of ResMPS to RNNs and Transformer Networks. Include performance of such networks on the benchmark problem if possible.

Reply: We added Sec. 2.5 titled "Comparision with the recurrent neural network and transformer" to discuss their relation. Please refer to major change 6.

  1. Include all details about the training procedure.

Reply: We added appendix A titled "Training details" to give more details, and codes are available on GitHub (see YemingMeng/ResMPS). Please refer to major changes 7&8.

  1. While the intermediate transformation of hidden features is well explained, it remained unclear to me how the final output that indicates the classification of the given data is produced by the network. This should be clarified.

Reply: We added the related details in the new version to improve the explanation (major change 3). The keypoint is that a linear layer without bias and activation implements this procedure.

  1. It would be helpful to initially clarify the domain of the training data x.

Reply: We also clarified this point in appendix A.

  1. For the ResMPS a one-dimensional ordering of the image data has to be chosen. How is this done? It could be interesting to investigate how the performance depends on the ordering.

Reply: The ordering we choose is called zigzag path, where pixels are arranged row by row. We added appendix B to compare different paths. The result shows that varying paths does not effect the accuracy significantly.

  1. At the end of section 2.2 you claim that the path of hidden features is similar for similar images. Then, you present a t-SNE for the final hidden features. Can you include a t-SNE of intermediate hidden features to prove the claim that already the paths are similar?

Reply: We added Fig. 3 to give more details.

  1. Regarding the factor of 2 between the parameter efficiency of MPS and sResMPS: Is this because the sResMPS fixes a gauge choice, which is left free in the MPS?

Reply: Yes. An alternative perspective is to investigate the 1st-order cutoff model, say $h^{[N]}-h^{[0]} \approx \sum_i\sum_{\alpha=1}^N x_{\alpha i} W^{[\alpha]}_i$, where $\alpha$ is the channel index. Doubling the channel number will double the terms on RHS, and hence LHS will also be doubled while applying the same perturbation on weights, and this will effectively double the learning rate.

  1. The discussion around Eq. (9) and (11) seems to be inconsistent with the definition of sResMPS in Eq. (4). In particular, how does the feature map enter Eq. (4)?

Reply: Actually, Eq. (9) and (10) is consistent with Eq. (4). We apologize that we omitted some intermediate formulas, so their equivalence doesn't seem so obvious. We added more derivation in the new version as Eq. (11) to Eq. (12), and this also answer the second question (major change 5).

  1. At the end of section 2.4 the expressivity of the ResMPS as function of $\chi$ and $M$ is discussed, coming to the conclusion that $\chi$ does not affect the expressivity of the ResMPS. However, only three large values of $\chi$ are considered and Fig. 3 shows that the expressivity is also not affected by varying $M$ over about one order of magnitude. How does the performance change when you vary $\chi$ between 1 and 20 ?

Reply: The statement "does not affect the expressibility of the ResMPS" is valid for sufficiently large $\chi$. If we verify $\chi$ from 2 to 12, the performance will increase at the start stage and achieves saturation after a critical point about $\chi_c=6$. The critical value $\chi_c$ is smaller than the number of classifications (major changes 11).

  1. The text contains a non-negligible amount of typos and grammatical errors. Please proof-read it thoroughly.

Reply: We apologize for the typos and grammatical errors and the inconvenience they have brought. We have proof-read it thoroughly and tried to revise each typos or grammatical errors that we have found. If any further problems, please let us know.

  1. Section 2.4 could be further divided into subsections for better clarity. In the current form, three different aspects with no immediate connection are discussed, which makes it hard to follow.

Reply: In the updated version, we divided the original section 2.4 into 3 subsections. After titling, we believe that the distinction and connection among the different parts of this section become clearer.

---

## Editorial Decision

resubmitted